# Early surgical treatment using regional clinical pathways to reduce the length of postoperative hospital stay in hip fracture patients: A retrospective analysis using the Japanese Diagnosis Procedure Combination database

Haruki Nishimura[1], Hitoshi Suzuki[1]*, Kei Tokutsu[2], Keiji Muramatsu[2], Makoto Kawasaki[1], Yoshiaki Yamanaka[1], Soshi Uchida[1], Eiichiro Nakamura[1], Kiyohide Fushimi[3], Shinya Matsuda[2], Akinori Sakai[1]

1 Department of Orthopaedics Surgery, School of Medicine, University of Occupational and Environmental Health, Kitakyushu, Japan, 2 Department of Preventive Medicine and Community Health, School of Medicine, University of Occupational and Environmental Health, Kitakyushu, Japan, 3 Department of Health Policy and Informatics, Tokyo Medical and Dental University Graduate School, Tokyo, Japan

* belltree@med.uoeh-u.ac.jp

## Abstract

Hip fracture is a common injury in older adults; however, the optimal timing of surgical treatment remains undetermined in Japan. Therefore, this retrospective study aimed to ascertain the rate of early surgery among hip fracture patients and investigate its effectiveness, along with "regional clinical pathways" (patient plan of care devised by Japanese clinicians), in reducing the length of hospital stay (LOS) postoperatively. We hypothesized that performing early surgery along with a regional clinical pathway is effective to reduce the postoperative LOS and complications among hip fracture patients. We examined the data of patients diagnosed with femoral neck and peritrochanteric fractures retrieved from the Japanese Diagnosis Procedure Combination database between April 2016 and March 2018. Patients were divided into the early (43,928, 34%; surgery within 2 days of admission) and delayed (84,237, 66%; surgery after 2 days of admission) surgery groups. The difference in postoperative LOS between the two groups was 3 days (early vs. delayed: 29 days vs. 32 days). The early surgery group had more cases of intertrochanteric fractures (57% vs. 43%) and internal fixation (74% vs. 55%) than did the delayed surgery group. In contrast, the delayed surgery group had more cases of femoral neck fractures (43% vs. 57%) and bipolar hip arthroplasty (25% vs. 42%) or total hip arthroplasty (1.2% vs. 3.0%). Moreover, the early surgery group showed a lower incidence of complications, except anemia (12% vs. 8.8%). Logistic regression analysis using the adjusted model revealed that early surgery and implementation of regional clinical pathways reduced LOS by 2.58 and 8.06 days, respectively (p<0.001). Early surgery and implementation of regional clinical pathways for hip fracture patients are effective in reducing postoperative LOS, allowing regional clinical pathways to

**Data Availability Statement:** All DPC datasets have ethical or legal restrictions for public deposition due to inclusion of sensitive information from the human subjects. All inquiries should be addressed to the Ethics Committee of Medical Care and Research of the University of Occupational and Environmental Health, Japan. The contact information of the Ethics Committee is as follows: 1-1 Iseigaoka, Yahatanishi-ku, Kitakyushu 807-8555, Japan; Tel +81-93-691-7205; E-mail daigakukanri@mbox.pub.uoeh-u.ac.jp.

**Funding:** This study was financially supported by the Japan Society for the Promotion of Science (JSPS) (https://www.jsps.go.jp/j-grantsinaid) in the form of a Grant-in-Aid for Scientific Research ((C) 23K08644) grant received by HS. No additional external funding was received for this study. The funder had no role in study design, data collection and analysis, decision to publish, or preparation of the manuscript.

**Competing interests:** The authors have declared that no competing interests exist.

have a greater impact. These findings will help acute care providers when treating hip fracture patients.

## Introduction

Hip fracture is common among older adults, especially those with osteoporosis. In Japan, approximately 200,000 cases of hip fracture were recorded in 2017 [1]. This number has been increasing worldwide, and it is estimated to increase globally to approximately seven million by 2050 [2]. Hip fracture affects not only the patient's ability to perform activities of daily living but also the long-term prognosis and quality of life. The proportion of older adults with a fractured hip who die within a year of injury is as high as 30%, while approximately 7% die during hospitalization [3–5]. Therefore, it is crucial to develop a concrete strategy for treating patients with hip fractures, especially patients at an advanced age.

The Japanese Orthopaedic Association has issued clinical guidelines for the treatment of hip fractures. However, there are no clear guidelines regarding the best surgical approach, especially regarding the optimal timing of surgery post-diagnosis. Current recommendations suggest that surgeons should perform surgery "as soon as possible" post-diagnosis. Conversely, the clinical guidelines of the American Association of Orthopaedic Surgeons and the National Health Service (UK) recommend performing surgery "within 48 hours of hospital admission". This recommendation is supported by the findings of previous studies that surgery for hip fracture delayed for longer than 48 hours may increase the risk of severe complications that may include surgical site infection, pressure sores, and other major medical complications, or even in-hospital and 1-year mortality [6–9]. However, some previous studies have reported no increase in complications or risk of 1-year mortality among patients who underwent surgical treatment delayed by more than 48 hours after hospital admission [10–12].

In Japan, more attention has been paid to implementation of regional clinical pathways to treat patients effectively and in a unified fashion. Regional clinical pathways are managed by outpatient clinics, acute care hospitals, rehabilitation hospitals, and care providers within a specific region to provide effective and consistent care. The protocol includes specifying the disease name, patient's symptoms at the time of hospitalization, hospital name, scheduled medical examinations and medical procedures, period until standard transfer, details of care post transfer, discharge criteria, standard time course for total care period, and other essential information [13, 14]. Regional clinical pathways also provide a series of support services, which include fracture treatment and discharge to home. This benefits both patients and hospitals by reducing turnaround time and enabling the efficient utilization of hospital beds.

In the present current study, we used data that were retrieved from the Diagnosis Procedure Combination (DPC) database, a Japanese nationally representative clinical database containing discharge abstract and administrative claim data. These include data regarding surgical procedures and medications indexed in the original Japanese codes assigned by the Ministry of Health, Labour and Welfare of Japan [15].

The objectives of this study were to i) determine the rate of early surgery among hip fracture patients, ii) investigate the effect of early surgery and regional clinical pathways on the length of postoperative hospital stay, and iii) use the DPC database to compare the postoperative complications of patients with hip fractures who underwent early surgery versus delayed surgery. We hypothesized that performing early surgery along with a regional clinical pathway is effective to reduce the postoperative length of hospital stay (LOS) and complications among hip fracture patients.

## Materials and methods

### Data sources

In this retrospective analysis, data retrieved from the Japanese DPC database between April 2016 and March 2018 were used. The study followed the principles of the Declaration of Helsinki and its later amendments. The protocol was approved by the Ethics Committee of Medical Care and Research of the University of Occupational and Environmental Health, Japan (approval no. R1-067). Due to the anonymized nature of the data, the requirement for informed consent was waived.

### Data collection

Patient data labelled with the codes S72.0 (femoral neck fracture) and S72.1 (femoral peritrochanteric fracture), according to the International Classification of Diseases (10th revision), were extracted from the DPC database.

We also retrieved the data of patients who underwent open reduction and internal fixation (ORIF), bipolar hip arthroplasty (BHA), and total hip arthroplasty (THA) after admission. The following patients were excluded: (1) those with missing data (n = 10,825) and (2) those with an LOS of >120 days (n = 1,271). Ultimately, we enrolled 128,165 patients in this study. Patients who underwent surgery within 2 days after admission were assigned to the early surgery group (43,928, 34%) and those who underwent surgery after more than 2 days post-admission were assigned to the delayed surgery group (84,237, 66%; Fig 1).

### Factor setting

First, the LOS data of all patients were collected and the multiple factors that could potentially affect postoperative LOS was ascertained. These factors included the following: patient details such as type of fracture, surgical procedure, sex, age, body mass index (BMI), comorbidities at admission such as diabetes mellitus, dementia, pneumonia, or osteoporosis, admission details such as ambulance transport, admission to a university hospital, application of a regional clinical pathway, transfer from another hospital. A regional clinical pathway is established to be shared and used in multiple coordinating medical institutions authorized to treat patients with health insurance coverage or nursing care facilities following patient transfer or discharge from the administering hospital. It records disease names, symptoms at the time of hospitalization, scheduled medical examinations, the average duration of patient transfers, contents of medical treatment after patient transfers, average time course of discharge from the coordinating medical institutions, release standard expected at the time of discharge, and other required items. Postoperative patient LOS-related factors included days until start of rehabilitation and complications such as anemia, need for blood transfusion, dislocation of the bipolar head or artificial hip joint, surgical site infection, deep vein thrombosis, pulmonary embolism, stroke, periprosthetic femoral fracture, pressure sore, pneumonia, delirium, and cardiovascular disease (ischemic heart disease, arrhythmias, heart failure, and hypertension).

### Statistical analysis

Each factor that could potentially affect postoperative LOS was compared as demographic data between the early surgery and delayed surgery groups. We then analyzed the contextual effects of early surgical treatment (within 2 days of admission) and hospital-level implementation of regional clinical pathways on LOS by employing multiple logistic regression analysis using the adjusted model. For this analysis, the prevalence of postoperative complications was used to adjust for background factors. In addition, the effects of sex, age, BMI, comorbidities at

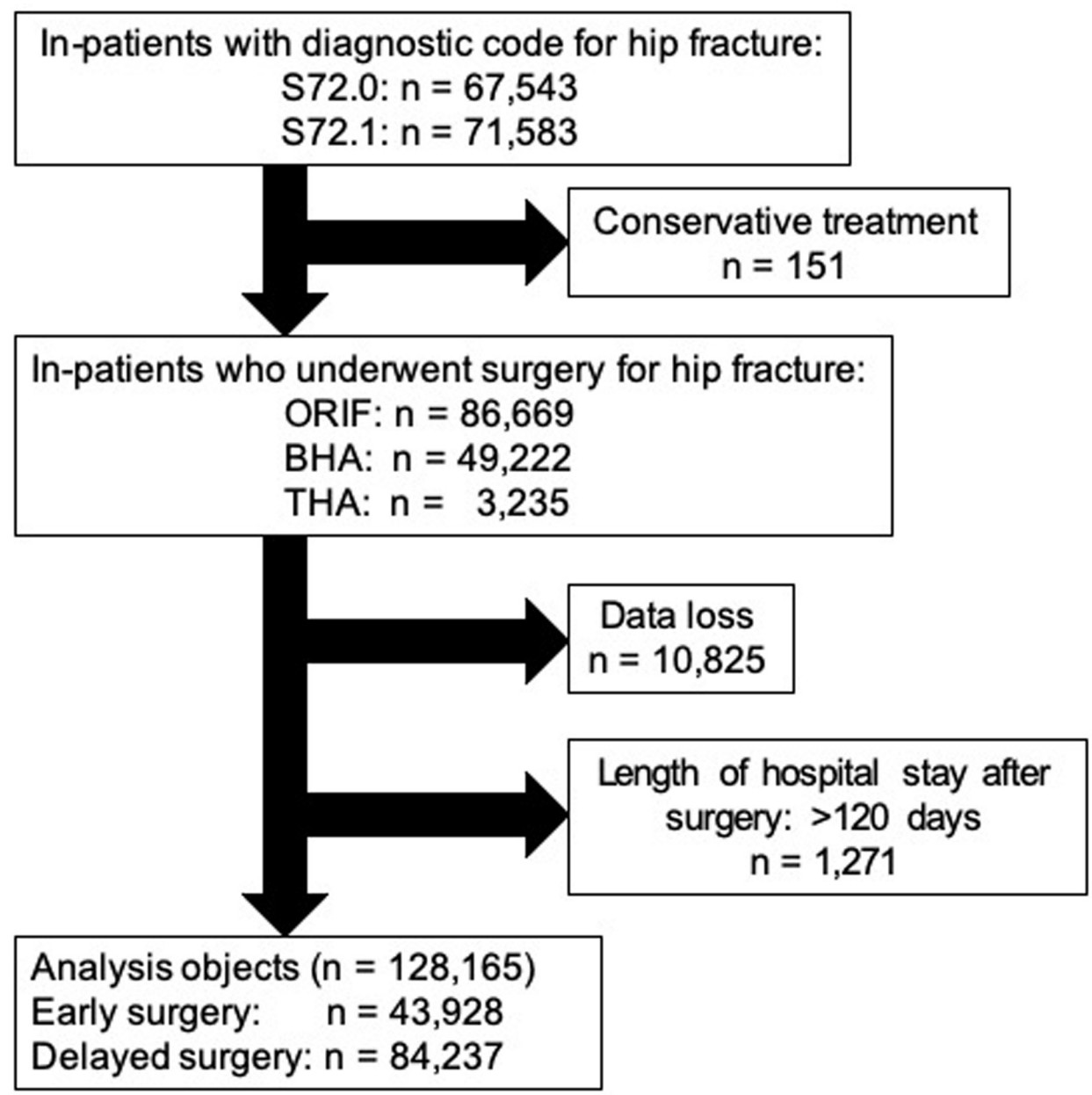

**Fig 1. Patient selection flowchart.** S72.0: femoral neck fracture and S72.1: femoral peritrochanteric fracture (codes from the International Classification of Diseases, 10th revision), ORIF: open reduction and internal fixation, BHA: bipolar hip arthroplasty, THA: total hip arthroplasty.

admission (diabetes mellitus, dementia, and pneumonia), admission to a university hospital, and transfer from another hospital on LOS were investigated. All calculations were performed using STATA Release 16.1 software (Stata, College Station, TX, USA).

## Results

### Characteristics of patients in the early and delayed surgery groups

The results of the distribution and comparison of patient characteristics between the early (surgery within 2 days) and delayed surgery groups (surgery after 2 days) are shown in

**Table 1. Distribution and comparison of patient characteristics between the early and delayed surgery groups.**

| Variables | Early surgery (n = 43,928; 34%) | Delayed surgery (n = 84,237; 66%) | P-value |
|---|---|---|---|
| **Total length of hospital stay, days (SD)** | 30.1 (22.5) | 37.4 (25.7) | <0.001 |
| **Length of stay after surgery, days (SD)** | 28.8 (21.7) | 32.1 (24.2) | <0.001 |
| **Diagnosis, yes, n (%)** | | | |
| Femoral neck fracture | 18,701 (43%) | 47,707 (57%) | <0.001 |
| Peritrochanteric fracture | 25,227 (57%) | 36,530 (43%) | <0.001 |
| **Surgical procedure, yes, n (%)** | | | |
| Open reduction and internal fixation | 32,660 (74%) | 46,682 (55%) | <0.001 |
| Bipolar hip arthroplasty | 10,756 (25%) | 35,038 (42%) | <0.001 |
| Total hip arthroplasty | 512 (1.2%) | 2,517 (3.0%) | <0.001 |
| **Sex, women, n (%)** | 34,029 (78%) | 64,198 (76%) | <0.001 |
| **Age, years, mean (SD)** | 83.4 (10.9) | 83.3 (9.7) | 0.51 |
| BMI, kg/m$^2$, mean (SD) | 20.57 (4.28) | 20.63 (4.53) | 0.033 |
| **Comorbidity at admission, yes, n (%)** | | | |
| Diabetes mellitus | 6,972 (16%) | 15,941 (19%) | <0.001 |
| Dementia | 1,722 (3.9%) | 3,499 (4.2%) | 0.044 |
| Pneumonia | 191 (0.4%) | 827 (1.0%) | <0.001 |
| Osteoporosis | 5,172 (12%) | 9,519 (11%) | 0.012 |
| **Ambulance transport, yes, n (%)** | 27,342 (62%) | 51,209 (61%) | <0.001 |
| **Admission to a university hospital, yes, n (%)** | 2,139 (4.5%) | 5,050 (5.5%) | <0.001 |
| **Use of regional clinical pathway, yes, n (%)** | 36,077 (76%) | 67,266 (74%) | <0.001 |
| **Transfer from another hospital on admission, yes, n (%)** | 19,906 (42%) | 36,831 (40%) | <0.001 |
| **Postoperative days until initiation of rehabilitation, days (SD)** | 1.61 (1.77) | 1.40 (1.35) | <0.001 |

n: sample size, SD: standard deviation

Table 1. The difference in total LOS in the hospital and postoperative LOS between the early surgery group and the delayed surgery group was 7 and 3 days, respectively (total LOS: 30.1 vs. 37.4 days [early vs. delayed], p<0.001; postoperative LOS: 28.8 vs. 32.1 days, p<0.001). The early surgery group had more cases of peritrochanteric fractures (57% vs. 43%, p<0.001) and internal fixation than did the delayed surgery group (74% vs. 55%, p<0.001). In contrast, the delayed surgery group had more cases of femoral neck fractures (43% vs. 57%, p<0.001) and BHA (25% vs. 42%, p<0.001) or THA (1.2% vs. 3.0%, p<0.001). The early surgery group showed a lower rate of comorbidities at admission than did the delayed surgery group, except for osteoporosis (diabetes, 16% vs. 19%, p<0.001; dementia, 3.9% vs. 4.2%, p = 0.044; pneumonia, 0.4% vs. 1.0%, p<0.001; osteoporosis, 12% vs. 11%, p = 0.012). In addition, the early surgery group used more ambulances at admission (62% vs. 61%, p<0.001). Interestingly, patients in the delayed surgery group experienced more hospitalizations in university hospitals than those in the early surgery group (4.5% vs 5.5%, p<0.001). In addition, the early surgery group used more regional clinical pathways and had more transfers from other hospitals (using a regional clinical pathway, 76% vs. 74%, p<0.001; transfer from another hospital, 42% vs. 40%, p<0.001).

## Postoperative complications in the early and delayed surgery groups

The details of postoperative complications are shown in Table 2. The rates of complications such as surgical site infection, deep vein thrombosis (DVT), pneumonia, and cardiovascular disease (p < 0.001) were significantly lower in the early surgery group than in the delayed

**Table 2. Postoperative complications.**

| Complications | Early surgery (n = 43,938; 34%) | Delayed surgery (n = 84,237; 66%) | P-value |
|---|---|---|---|
| **Anemia** | 5,255 (12%) | 7,410 (8.8%) | <0.001 |
| **Blood transfusion** | 18,264 (42%) | 31,634 (38%) | <0.001 |
| **Bipolar head or artificial hip joint dislocation** | 60 (0.1%) | 132 (0.2%) | 0.4 |
| **Surgical site infection** | 356 (0.8%) | 834 (1.0%) | 0.001 |
| **Deep vein thrombosis** | 1,649 (3.8%) | 3,832 (4.5%) | <0.001 |
| **Pulmonary embolism** | 94 (0.2%) | 275 (0.3%) | <0.001 |
| **Stroke** | 271 (0.6%) | 524 (0.6%) | 0.94 |
| **Periprosthetic femoral fracture** | 38 (0.1%) | 102 (0.1%) | 0.09 |
| **Pressure sore** | 38 (0.1%) | 102 (0.1%) | 0.09 |
| **Pneumonia** | 416 (0.9%) | 1,076 (1.3%) | <0.001 |
| **Delirium** | 987 (2.2%) | 1,863 (2.2%) | 0.69 |
| **Cardiovascular disease*** | 1,652 (3.8%) | 4,072 (4.8%) | <0.001 |

n: sample size

*Cardiovascular disease includes ischemic heart disease, arrhythmias, heart failure, and hypertension.

surgery group. However, the incidence rates of anemia and blood transfusion were significantly higher in the early surgery group than in the delayed surgery group (anemia: 12% vs. 8.8%, p < 0.001; blood transfusion: 42% vs. 38%, p < 0.001).

## Factors associated with LOS among hip fracture patients

The results on the association of variables with postoperative LOS obtained from logistic regression analysis using the adjusted model are shown in Table 3. Factors such as comorbidity at admission (diabetes mellitus and pneumonia), admission to a university hospital, and transfer from another hospital at admission were significantly associated with postoperative LOS. After adjusting for these factors, early surgery and application of a regional clinical pathway significantly reduced the postoperative LOS by 2.58 (p < 0.001) and 8.06 (p < 0.001) days, respectively. Furthermore, admission to a university hospital and transfer from another

**Table 3. Multilevel regression analysis of length of stay after surgery.**

| Variables | Coefficient | SE | 95% CI | | P-value |
|---|---|---|---|---|---|
| **Surgery within 2 days** | -2.58 | 0.12 | -2.81 | -2.34 | <0.001 |
| **Using a regional clinical pathway** | -8.06 | 0.13 | -8.32 | -7.80 | <0.001 |
| **Sex, women** | -0.44 | 0.14 | -0.71 | -0.17 | 0.001 |
| **Age** | 0.12 | 0.01 | 0.11 | 0.13 | <0.001 |
| **BMI, kg/m² (vs ≥18.5, <25)** | 0.06 | 0.01 | 0.04 | 0.09 | <0.001 |
| **Comorbidity** | | | | | |
| Diabetes mellitus | 1.67 | 0.15 | 1.37 | 1.96 | <0.001 |
| Dementia | -1.18 | 0.29 | -1.74 | -0.61 | <0.001 |
| Pneumonia | 0.48 | 0.64 | -0.79 | 1.74 | 0.458 |
| **Admission to university hospital** | -5.68 | 0.26 | -6.19 | -5.17 | <0.001 |
| **Transfer from another hospital at admission** | -5.00 | 0.12 | -5.22 | -4.77 | <0.001 |

SE: standard error, CI: confidence interval

hospital at admission significantly reduced the postoperative LOS by 5.68 and 5.00 (p < 0.001), respectively.

## Discussion

We examined the efficacy of early surgery and implementation of regional clinical pathways in reducing the length of postoperative hospital stay. We found that early surgery within 2 days of hospitalization, which is prevalent in one-third of all hip fracture patients, was effective in reducing postoperative LOS by approximately 3 days. Furthermore, the early surgery group showed a significantly lower incidence of most postoperative complications than did the delayed surgery group. However, compared with patients in the delayed surgery group, those in the early surgery group were more frequently diagnosed with peritrochanteric fracture and underwent ORIF. Notably, our results showed that a regional clinical pathway to treat such patients had a greater advantage in reducing LOS than did early surgical treatment. Early surgical treatment and use of regional clinical pathways decreased the postoperative LOS by 2.58 and 8.06 days, respectively.

### Difference in patient characteristics between the early and delayed surgery group

Our data showed that the early surgery group had fewer comorbidities upon admission than did the delayed surgery group. Comparing the two groups, patients in the early surgery group were diagnosed with peritrochanteric fracture and underwent ORIF more frequently, whereas patients in the delayed surgery group were diagnosed with femoral neck fracture and underwent BHA or THA more frequently. However, it was surprising that the number of patients in the delayed surgery group was twice the number of patients in the early surgery group, even though it is well known that early surgery is preferable. Since internal fixation for intertrochanteric femur fractures is less invasive and has a better prognosis than do BHA and THA [16], it is believed that the surgery was planned early after the injury. However, it is sometimes difficult to perform early surgery in hip fracture patients, given their comorbidities and poor general condition. A previous study investigating the impact of factors contributing to delayed hip fracture surgery on 1-year mortality rates revealed that patients with logistical concerns (such as unavailability of operating rooms and surgeons or insufficient time in the operating room, as well as admission on weekends or holidays) showed a significantly higher incidence of postoperative complications and a higher 1-year mortality rate than did patients without these concerns [17]. Thus, it was recommended that the type and timing of surgery should be determined based on the general condition of each patient. Logistical concerns should not become the reason for delaying surgery.

### Difference in postoperative complications between the early and delayed surgery groups

In our study, the early surgery group showed remarkably fewer postoperative complications, except anemia, than did the delayed surgery group. This result was consistent with those of previous studies that investigated the association between surgery conducted within 48 hours and postoperative complications among hip fracture patients [18–20]. There is a consensus that early surgery for hip fracture can reduce most postoperative complications and even mortality [21]. A lower postoperative incidence of complications contributes to shortened LOS after surgery. Along with LOS, a waiting time of more than 24 hours for hip fracture surgery is also associated with an increase in medical costs [22].

As mentioned earlier, the early surgery group in this study showed a remarkably higher incidence of anemia and blood transfusion requirement. To the best of our knowledge, no previous study has compared the incidence of anemia and blood transfusion requirement between the early and delayed surgery groups among hip fracture patients. A large amount of blood loss among hip fracture patients often occurs immediately after injury and before surgery. The average quantity of blood loss has been reported to be greater with peritrochanteric fractures than with femoral neck fractures [23]. Therefore, the early surgery group can be assumed to have more cases of anemia and blood transfusion. Bian et al. [24] investigated the influence of preoperative risk factors on postoperative blood transfusion after hip fracture surgery and revealed that older age, general anesthesia, low preoperative hemoglobin, and non-use of tranexamic acid were independently involved. Therefore, clinicians should be aware of the optimal use of blood transfusion for hip fracture patients, depending on patients' general condition and potential risk factors.

In this study, the delayed surgical group showed a significantly higher incidence of DVT. Song et al. [25] reported that delayed hip fracture surgery was a risk factor for preoperative DVT, and most patients who were diagnosed with DVT after surgery had already developed a thrombus before surgery. Zhang et al. [26] investigated the incidence of DVT in the lower extremities before and after hip fracture surgery. They reported that the number of days between fracture and operation, amount of blood loss, development of venous thrombosis at admission, and presence of coronary heart disease were independent risk factors of postoperative DVT. Fu et al. [27] conducted a similar study and reported that compared with BHA and THA, blood loss was an independent risk factor, while ORIF was an independent protective factor for postoperative DVT. Although the early surgery group in our study showed a higher incidence of anemia than did the delayed surgery group, the findings of these studies are consistent with our assumptions.

## Contribution of early surgery to LOS reduction

Our results showed that early surgery within 2 days of admission shortened the postoperative LOS by 2.58 days and reduced most of the postoperative complications, except anemia and blood transfusion. Therefore, early surgery for hip fracture patients within 2 days is highly recommended. Additionally, concomitant diabetes mellitus or pneumonia increased LOS, suggesting that the preoperative management of these comorbidities is crucial. Lizaur-Utrilla et al. [17] suggested that a waiting time of more than 2 days was not associated with a higher mortality rate or with the development of complications if the time was used to stabilize active comorbidities in patients upon admission.

## Contribution of regional clinical pathways and other factors to LOS reduction

Our results demonstrated that implementation of regional clinical pathways reduced the postoperative LOS for hip fracture patients by 8.06 days. Takahashi et al. [28] investigated the factors associated with LOS after hip fracture surgery and reported that admission to a high-volume surgery hospital contributed to LOS reduction. Although no previous study investigated the relationship between LOS and the implementation of regional clinical pathways among hip fracture patients, Fujino et al. [29] investigated the effect of regional clinical pathways on LOS in stroke patients; the use of regional clinical pathways at the hospital and patient levels reduced LOS by 9.1 days and 7.2 days, respectively. Therefore, sharing a unified postoperative protocol by implementing regional clinical pathways leads to effective utilization of

hospital beds. In summary, use of a clinical pathway can facilitate discharging of postoperative patients, thus reducing the postoperative LOS.

Transfer from another hospital at admission also reduced the postoperative LOS for hip fracture patients by 5.00 days. We speculate that this was possible because of good cooperation between small hospitals that cannot perform surgeries and big hospitals that perform surgery. In general, they cooperate well with each other to facilitate smooth transfers and surgeries for patients in Japan. Therefore, the early surgery group had more patients transferred from another hospital at admission, and this factor effectively reduced the postoperative LOS.

## Limitations

This study has some limitations. First, since we utilized the DPC database for this study, our data may not present sufficient information for all patients with hip fractures. In addition, the DPC database only includes the data of patients from acute care hospitals and this may have caused an inadvertent selection bias. Therefore, patient data post-discharge were not obtained. In addition, there was a significant difference between the early surgery and delayed surgery groups in terms of comorbidities at admission, and patients with fewer comorbidities on admission could have received earlier surgery; thus, there was a possibility of selection bias. Furthermore, we only compared early and delayed surgery groups and did not analyze the data using absolute timeframes, particularly for the delayed surgery group. Future studies are warranted to assess the impact of early surgical treatment and robust implementation of regional clinical pathways throughout the treatment of hip fracture patients.

## Conclusion

This study revealed that early surgery for hip fracture patients (within 2 days after admission) reduced postoperative complications and postoperative LOS by 2.6 days. Moreover, the implementation of regional clinical pathways to treat these patients had a greater impact on postoperative LOS, reducing it by 8.1 days. On the basis of these results, early surgery with implementation of a regional clinical pathway is strongly recommended for the treatment for the hip fracture patients. These study findings will help acute healthcare providers attend to hip fracture patient needs more efficiently and to reduce admission processing time as well as the hospital LOS, which are factors that can be psychologically and financially taxing, especially for older patients.

## Acknowledgments

We would like to thank Editage (www.editage.com) for English language editing.

## Author Contributions

**Conceptualization:** Haruki Nishimura, Hitoshi Suzuki, Kei Tokutsu, Keiji Muramatsu, Makoto Kawasaki, Yoshiaki Yamanaka, Soshi Uchida, Eiichiro Nakamura, Kiyohide Fushimi, Akinori Sakai.

**Data curation:** Haruki Nishimura, Kei Tokutsu, Keiji Muramatsu, Kiyohide Fushimi, Shinya Matsuda.

**Formal analysis:** Kei Tokutsu, Keiji Muramatsu, Kiyohide Fushimi.

**Funding acquisition:** Hitoshi Suzuki.

**Investigation:** Haruki Nishimura, Hitoshi Suzuki, Kei Tokutsu, Keiji Muramatsu, Yoshiaki Yamanaka, Kiyohide Fushimi.

**Methodology:** Kei Tokutsu, Keiji Muramatsu, Kiyohide Fushimi, Shinya Matsuda, Akinori Sakai.

**Project administration:** Hitoshi Suzuki, Kei Tokutsu, Keiji Muramatsu.

**Resources:** Kei Tokutsu, Keiji Muramatsu, Kiyohide Fushimi, Shinya Matsuda.

**Software:** Kei Tokutsu, Keiji Muramatsu, Shinya Matsuda.

**Supervision:** Hitoshi Suzuki, Makoto Kawasaki, Yoshiaki Yamanaka, Soshi Uchida, Eiichiro Nakamura, Kiyohide Fushimi, Shinya Matsuda, Akinori Sakai.

**Validation:** Hitoshi Suzuki, Kei Tokutsu, Keiji Muramatsu, Kiyohide Fushimi, Shinya Matsuda.

**Visualization:** Haruki Nishimura, Kei Tokutsu, Makoto Kawasaki, Yoshiaki Yamanaka, Soshi Uchida, Eiichiro Nakamura, Akinori Sakai.

**Writing – original draft:** Haruki Nishimura.

**Writing – review & editing:** Haruki Nishimura, Hitoshi Suzuki, Kei Tokutsu, Keiji Muramatsu, Makoto Kawasaki, Yoshiaki Yamanaka, Soshi Uchida, Eiichiro Nakamura, Akinori Sakai.

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
