## [Decision Letter · Decision Letter 0]

31 Oct 2023

PONE-D-23-05138Early surgical treatment using regional clinical pathways to reduce the length of postoperative hospital stay in hip fracture patients: A retrospective analysis using the Japanese Diagnosis Procedure Combination databasePLOS ONE

Dear Dr. Suzuki,

Thank you for submitting your manuscript to PLOS ONE. After careful consideration, we feel that it has merit but does not fully meet PLOS ONE’s publication criteria as it currently stands. Therefore, we invite you to submit a revised version of the manuscript that addresses the points raised during the review process.

Additionally, the reviewers and I want to thank you for your submission to PLOS One. We recognize that you have a multitude of journals to which you could submit your work. So thank you for choosing PLOS One. Second, I want to apologize for the significant delay in reviewing your manuscript. It has taken an extraordinary amount of time for us to locate appropriate and willing reviewers for this manuscript. So thank you for your patience while we reviewed the manuscript. We have now received the necessary number of reviews for your manuscript in addition to me personally reviewing your submission. Please see below for comments from the review and editorial staff: **Reviewer #1**I had the honor to review this great paper titled “Early surgical treatment using regional clinical pathways to reduce the length of postoperative hospital stays in hip fracture patients: A retrospective analysis using the Japanese Diagnosis Procedure Combination database”. I think it consumed a lot of effort and time to achieve this paper. I have few comments.1- Introduction section:Line 51-53: these statistics belong to which country?? Even the reference refers to Japanese statistics, the text itself is not clear.2-Results section:The prevalence of preexisting comorbidities is strongly affecting the results especially the length of stay LOS, so attributing this reduction of LOS to only the implementation of regional clinical pathways is not accurate, this is a selection bias. Even so, in the limitation section, no referral to this point, thus both early and delayed surgery groups are not the same in prevalence of comorbidities which typically will influence the results. So, I want a clear statement about this point.Conclusion section:I think should be rephrased, with clearer strong statement, with rearrangement of sentencesFirst sentences, lines 300~303 just describe a fact which is not discovered by the authors nor first reported. I suggest starting the second sentence; i., e about relation between early surgery and rate of complication should be the striking finding that make the clear impression with the reader.Thank you very much for your efforts.

**Reviewer #2**

This is an interesting study analyzing effects of surgical delay and regional clinical pathways on the length of hospital stay in hip fracture patients utilizing a Japanese database. The authors should be commended for their effort. However, this study needs significant revisions to be considered for publication.The manuscript should focus on the effect of regional clinical pathways on LOS. See below for more information.Title = OkAbstractHypothesis is missing.IntroductionHypothesis is missing.MethodsPlease explain the regional pathways.Results & DiscussionThere are more comorbidities in the delayed group. Thus, the increased postoperative complication rate is not surprising.Table 2 – What is meant by cardiovascular disease as a complication? Heart attack?The authors need to clarify the regional pathways. The 3-week length of stay is long and not typical in other countries such as the USA. Further, it is unclear to me how the LOS of a transferred patient is shorter. Are they being transferred the same day?The data needs to be analyzed accounting for delayed surgery not only for more than 48h but also for the absolute timeframe. It is not surprising that the LOS of a patient who has surgery in 5 days may be longer than the one in 3 days.There is a lower rate of anemia in the delayed group which is surprising as more blood loss would be expected with an untreated femur fracture. The discussion should provide a better explanation.The main result of this study is the effect of regional clinical pathways on LOS. This needs to be explained more detail and should be the main focus of this manuscript. What are these pathways? Are they different from international pathways?Figures and TablesTable 1 – There is a significant difference between postoperative days until rehabilitation initiated although both are listed as one day. Please explain and add decimals.References = Adequate.Please submit your revised manuscript by December 1st, 2023. If you will need more time than this to complete your revisions, please reply to this message or contact the journal office at plosone@plos.org. Please include the following items when submitting your revised manuscript:A rebuttal letter that responds to each point raised by the academic editor and reviewer(s). You should upload this letter as a separate file labeled 'Response to Reviewers'.A marked-up copy of your manuscript that highlights changes made to the original version. You should upload this as a separate file labeled 'Revised Manuscript with Track Changes'.An unmarked version of your revised paper without tracked changes. You should upload this as a separate file labeled 'Manuscript'.

We look forward to receiving your revised manuscript.

Kind regards,

Ryan Chapman, PhD

Academic Editor

PLOS ONE

Journal Requirements:

Reviewers' comments:

Reviewer's Responses to Questions

**Comments to the Author**

1. Is the manuscript technically sound, and do the data support the conclusions?

Reviewer #1: Yes

Reviewer #2: Partly

2. Has the statistical analysis been performed appropriately and rigorously? 

Reviewer #1: Yes

Reviewer #2: Yes

3. Have the authors made all data underlying the findings in their manuscript fully available?

Reviewer #1: No

Reviewer #2: No

4. Is the manuscript presented in an intelligible fashion and written in standard English?

Reviewer #1: Yes

Reviewer #2: Yes

5. Review Comments to the Author

Reviewer #1: Dear respected Editor en Chief,

Dear respected Authors.

I had the honor to review this great paper titled “Early surgical treatment using regional clinical pathways to reduce the length of

postoperative hospital stays in hip fracture patients: A retrospective analysis using the Japanese Diagnosis Procedure Combination database”. I think it consumed a lot of effort and time to achieve this paper. I have few comments.

1- Introduction section:

Line 51-53: these statistics belong to which country?? Even the reference refers to Japanese statistics, the text itself is not clear.

2-Results section:

The prevalence of preexisting comorbidities is strongly affecting the results especially the length of stay LOS, so attributing this reduction of LOS to only the implementation of regional clinical pathways is not accurate, this is a selection bias. Even so, in the limitation section, no referral to this point, thus both early and delayed surgery groups are not the same in prevalence of comorbidities which typically will influence the results. So, I want a clear statement about this point.

Conclusion section:

I think should be rephrased, with clearer strong statement, with rearrangement of sentences.

First sentences, lines 300~303 just describe a fact which is not discovered by the authors nor first reported. I suggest starting the second sentence; i., e about relation between early surgery and rate of complication should be the striking finding that make the clear impression with the reader.

Thank you very much for your efforts.

Reviewer #2: This is an interesting study analyzing effects of surgical delay and regional clinical pathways on the length of hospital stay in hip fracture patients utilizing a Japanese database. The authors should be commended for their effort. However, this study needs significant revisions to be considered for publication.

The manuscript should focus on the effect of regional clinical pathways on LOS. See below for more information.

Title.

Ok

Abstract

Hypothesis is missing.

Introduction

Hypothesis is missing.

Methods

Please explain the regional pathways.

Results & Discussion

There are more comorbidities in the delayed group. Thus, the increased postoperative complication rate is not surprising.

Table 2 – What is meant by cardiovascular disease as a complication? Heart attack?

The authors need to clarify the regional pathways. The 3-week length of stay is long and not typical in other countries such as the USA. Further, it is unclear to me how the LOS of a transferred patient is shorter. Are they being transferred the same day?

The data needs to be analyzed accounting for delayed surgery not only for more than 48h but also for the absolute timeframe. It is not surprising that the LOS of a patient who has surgery in 5 days may be longer than the one in 3 days.

There is a lower rate of anemia in the delayed group which is surprising as more blood loss would be expected with an untreated femur fracture. The discussion should provide a better explanation.

The main result of this study is the effect of regional clinical pathways on LOS. This needs to be explained more detail and should be the main focus of this manuscript. What are these pathways? Are they different from international pathways?

Figures and Tables

Table 1 – There is a significant difference between postoperative days until rehabilitation initiated although both are listed as one day. Please explain and add decimals.

References

Adequate.

6. PLOS authors have the option to publish the peer review history of their article (what does this mean?). If published, this will include your full peer review and any attached files.

Reviewer #1: **Yes: **Ahmed Hamed Kassem Abdelaal

Reviewer #2: No

---

## [Author Response · Author response to Decision Letter 0]

12 Dec 2023

Reviewer 1

1. Introduction section:

Line 51-53: these statistics belong to which country?? Even the reference refers to Japanese statistics, the text itself is not clear.

Response: We apologize for the lack of clarity. The statistics are for Japan; this has now been specified in the revised manuscript (Line 53).

2. Results section:

The prevalence of preexisting comorbidities is strongly affecting the results especially the length of stay LOS,　so attributing this reduction of LOS to only the implementation of regional clinical pathways is not accurate, this is a selection bias. Even so, in the limitation section, no referral to this point, thus both early and delayed surgery groups are not the same in prevalence of comorbidities which typically will influence the results. So, I want a clear statement about this point.

Response: Thank you for your valuable insights. We expected that comorbidities at admission would affect the length of hospital stay. In fact, Table 1 shows that the early surgery group had fewer comorbidities on admission than did the delayed surgery group. On the other hand, Table 3 shows the effect of early surgery and the regional clinical pathway on the length of hospital stay after adjusting for admission comorbidities and postoperative complications as background factors. However, as you pointed out, there may have been a selection bias because patients with fewer comorbidities on admission could have received earlier surgery. We have included this point as a limitation of the study in the revised manuscript (Lines 318-322).

3. Conclusion section:

I think should be rephrased, with clearer strong statement, with rearrangement of sentences. First sentences, lines 300~303 just describe a fact which is not discovered by the authors nor first reported. I suggest starting the second sentence; i., e about relation between early surgery and rate of complication should be the striking finding that make the clear impression with the reader.

Response: Thank you for your valuable suggestions. We have modified the conclusion section per your advice (Lines 327-335).

Reviewer 2

1. Abstract:

Hypothesis is missing

Response: We apologize for the missing hypothesis. It has now been added in the Abstract as follows: 

“We hypothesized that early surgery with implementation of a regional clinical pathway would be effective in reducing postoperative LOS and complications in hip fracture patients.” (Lines 30-32)

2. Introduction

Hypothesis is missing.

Response: We apologize for the missing hypothesis. It has now been added in the Introduction (Lines 94-96).

3. Methods:

Please explain the regional pathways.

Response: Thank you for your advice. An explanation of a reginal clinical pathway has been added in the Materials and Methods section (Factor setting subsection) as follows:

“A regional clinical pathway is established to be shared and used in multiple coordinating medical institutions authorized to treat patients with health insurance coverage or nursing care facilities following patient transfer or discharge from the administering hospital. It records disease names, symptoms at the time of hospitalization, scheduled medical examinations, the average duration of patient transfers, contents of medical treatment after patient transfers, average time course of discharge from the coordinating medical institutions, release standard expected at the time of discharge, and other required items.” (Lines 131-138).

4. There are more comorbidities in the delayed group. Thus, the increased postoperative complication rate is not surprising.

Response: Thank you for pointing this out. The prevalence of postoperative complications in the delayed surgery group was as expected, and it was used to adjust for background factors when identifying the effect on the postoperative hospital stay, as described in Table 3.

This is also true for variables such as comorbidities at admission in Table 1, which were also used to adjust for background factors. In the Statistical analysis section, we have specified that the prevalence of postoperative complications was used to adjust for background factors in the multiple logistic regression analysis (Lines 151-152).

5. Table 2 – What is meant by cardiovascular disease as a complication? Heart attack?

Response: We apologize for the lack of clarity. Cardiovascular disease included ischemic heart disease, arrhythmias, heart failure, and hypertension. We have specified this in the Materials and Methods section (Factor setting subsection) and the legend for Table 2 (Lines 143-144 and 195-196).

6. The authors need to clarify the regional pathways. The 3-week length of stay is long and not typical in other countries such as the USA. Further, it is unclear to me how the LOS of a transferred patient is shorter. Are they being transferred the same day?

Response: We apologize for the lack of clarity. First, we have added a description of the regional clinical pathway in the Materials and Methods section (Factor setting subsection) (Lines 131-138).

Second, a recent study reported that the average length of hospital stay for hip fracture patients in Japan is 31 days (Maki et al, Impact of the Comorbidity Polypharmacy Score on Clinical Outcome in Patients with Hip Fracture undergoing surgery Using Real-World Data, Annals of Clinical Epidemiology, 2023, 5, 3, p. 88-94,). Therefore, a total LOS of 30-37 days, as documented in the present study, is reasonable. 

Although we do not have data to show whether or not the transferred patients were transferred to the hospital for surgery on the same day, we believe that most of them were transferred on the same day or 1 day after their injury. We speculate that the reduced LOS for transferred patients was possible because of good cooperation between small hospitals that cannot perform surgeries and big hospitals that perform surgery. In general, they cooperate well with each other to facilitate smooth transfers and surgeries for patients in Japan. Therefore, the early surgery group had more patients transferred from another hospital at admission, and this factor effectively reduced the postoperative LOS. We have added this explanation in the Discussion section (Lines 305-311).

7. The data needs to be analyzed accounting for delayed surgery not only for more than 48h but also for the absolute timeframe. It is not surprising that the LOS of a patient who has surgery in 5 days may be longer than the one in 3 days.

Response: Thank you for your valuable insights. In this study, the effect of early surgery on postoperative hospital stay was calculated using logistic regression analysis with an adjusted model. The results showed that early surgery within 2 days was effective in reducing postoperative LOS by 2.58 days, which is considered to be a reduction in LOS excluding the waiting period for surgery. On the other hand, we could not analyzed the effect on LOS using an absolute timeframe including the waiting period. This has been mentioned as a limitation of the study in the Discussion section (Lines 321-322).

8. There is a lower rate of anemia in the delayed group which is surprising as more blood loss would be expected with an untreated femur fracture. The discussion should provide a better explanation.

Response: Thank you for your comments. This result is attributed to the fact that the early surgery group included more peritrochanteric fractures, which result in greater blood loss than do femoral neck fractures, which were more common in the delayed surgery group. We have discussed this point in the Discussion section (Lines 256-263).

9. The main result of this study is the effect of regional clinical pathways on LOS. This needs to be explained more detail and should be the main focus of this manuscript. What are these pathways? Are they different from international pathways?

Response: Thank you for your comments. We have added a subsection titled “Contribution of regional clinical pathways and other factors to LOS reduction” in the Discussion section in order to emphasize the effect of regional clinical pathways on LOS reduction (Lines 292-304). 

We have also added a description of the regional clinical pathway in the Materials and Methods section (Factor setting subsection) (Lines 131-138).

10. Table 1 – There is a significant difference between postoperative days until rehabilitation initiated although both are listed as one day. Please explain and add decimals.

Response: Thank you for pointing this out. We apologize for the error; these values were the same because of an error in the statistical software settings. We have recalculated and corrected the values. Postoperative days until initiation of rehabilitation were 1.61 and 1.40 days in the early and delayed surgery groups, respectively. The table has been rectified. 

Note:

 We kindly ask PLOS ONE to update the grant information in Financial Disclosure as following:

Currently- HS received a Grant-in-Aid for Scientific Research [grant numbers (C) 20K09421] from the Japan Society for the Promotion of Science (JSPS). https://www.jsps.go.jp/j-grantsinaid/ The funders had no role in study design, data collection and analysis, decision to publish, or preparation of the manuscript.

Update- HS received a Grant-in-Aid for Scientific Research [grant numbers (C) 20K09421 and (C) 23K08644] from the Japan Society for the Promotion of Science (JSPS). https://www.jsps.go.jp/j-grantsinaid/ The funders had no role in study design, data collection and analysis, decision to publish, or preparation of the manuscript.

---

## [Decision Letter · Decision Letter 1]

5 Jun 2024

Early surgical treatment using regional clinical pathways to reduce the length of postoperative hospital stay in hip fracture patients: A retrospective analysis using the Japanese Diagnosis Procedure Combination database

PONE-D-23-05138R1

Dear Dr. Suzuki,

We’re pleased to inform you that your manuscript has been judged scientifically suitable for publication and will be formally accepted for publication once it meets all outstanding technical requirements.

Kind regards,

Nan Jiang

Academic Editor

PLOS ONE

Additional Editor Comments (optional):

Reviewers' comments:

Reviewer's Responses to Questions

**Comments to the Author**

1. If the authors have adequately addressed your comments raised in a previous round of review and you feel that this manuscript is now acceptable for publication, you may indicate that here to bypass the “Comments to the Author” section, enter your conflict of interest statement in the “Confidential to Editor” section, and submit your "Accept" recommendation.

Reviewer #3: All comments have been addressed

2. Is the manuscript technically sound, and do the data support the conclusions?

Reviewer #3: Yes

3. Has the statistical analysis been performed appropriately and rigorously? 

Reviewer #3: Yes

4. Have the authors made all data underlying the findings in their manuscript fully available?

Reviewer #3: Yes

5. Is the manuscript presented in an intelligible fashion and written in standard English?

Reviewer #3: Yes

6. Review Comments to the Author

Reviewer #3: The authors have adequately addressed all the comments raised by previous reviewers. So, the manuscript now deserves the credit for publication.

7. PLOS authors have the option to publish the peer review history of their article (what does this mean?). If published, this will include your full peer review and any attached files.

Reviewer #3: **Yes: **Sreenivasulu Metikala

---

## [Editor Report · Acceptance letter]

22 Jul 2024

PONE-D-23-05138R1 

PLOS ONE

Dear Dr. Suzuki, 

I'm pleased to inform you that your manuscript has been deemed suitable for publication in PLOS ONE. Congratulations! Your manuscript is now being handed over to our production team.

Kind regards, 

on behalf of

Dr. Nan Jiang 

Academic Editor

PLOS ONE